# The Distinctive Features behind the Aggressiveness of Oral and Cutaneous Squamous Cell Carcinomas

**DOI:** 10.3390/cancers15123227

**Published:** 2023-06-17

**Authors:** Miguel Alonso-Juarranz, Marta Mascaraque, Elisa Carrasco, Tamara Gracia-Cazaña, Oscar De La Sen, Yolanda Gilaberte, Salvador Gonzalez, Ángeles Juarranz, Farzin Falahat

**Affiliations:** 1Oral and Maxillofacial Surgery Service, Hospital Clínico San Carlos, 28040 Madrid, Spain; mialon09@ucm.es (M.A.-J.); odelasen@ucm.es (O.D.L.S.); 2Surgery Department, Faculty of Medicine, Universidad Complutense, 28040 Madrid, Spain; 3Department of Biology, Universidad Autónoma de Madrid, 28049 Madrid, Spain; marta.mascaraque@uam.es (M.M.); elisa.carrasco@uam.es (E.C.); 4Translational Research Unit, Miguel Servet University Hospital, Instituto Investigación Sanitaria Aragón (IIS), 50009 Zaragoza, Spain; 5Department of Experimental Dermatology and Skin Biology, Instituto Ramón y Cajal de Investigación Sanitaria, IRYCIS, 28034 Madrid, Spain; salvagonrod@gmail.com; 6Department of Dermatology, Miguel Servet University Hospital, Instituto Investigación Sanitaria Aragón (IIS), 50009 Zaragoza, Spain; tamgracaz@gmail.com (T.G.-C.); ygilaberte@gmail.com (Y.G.); 7Department of Medicine and Medical Specialties, Universidad de Alcalá, 28871 Madrid, Spain

**Keywords:** oral squamous cell carcinoma, cutaneous squamous cell carcinoma, tumor microenvironment, cancer-associated fibroblasts, immune cells, tertiary lymphoid structures

## Abstract

**Simple Summary:**

In this review, we describe the recent studies that define the genetic alterations and composition of the stroma of oral and cutaneous squamous cell carcinomas (OSCC and CSCC, respectively). Mutations in tumor suppressor genes and protooncogenes cooperate in determining the differentiation, aggressiveness, and metastatic potential of these types of cancers. Driver mutations in tumor suppressor genes are more frequently observed in OSCC than CSCC. We also describe the differential composition of the tumor microenvironment and how this influences the aggressiveness of each tumor type. Although both OSCC and CSCC tumors are highly infiltrated by immune cells, high levels of tumor-infiltrating lymphocytes have been more frequently reported as predictors of good responses in OSCC than CSCC.

**Abstract:**

Squamous cell carcinomas arise from stratified squamous epithelia. Here, a comparative analysis based on recent studies defining the genetic alterations and composition of the stroma of oral and cutaneous squamous cell carcinomas (OSCC and CSCC, respectively) was performed. Both carcinomas share some but not all histological and genetic features. This review was focused on how mutations in tumor suppressor genes and protooncogenes cooperate to determine the differentiation, aggressiveness, and metastatic potential of OSCC and CSCC. In fact, driver mutations in tumor suppressor genes are more frequently observed in OSCC than CSCC. These include mutations in *TP53* (encoding pP53 protein), *CDKN2A* (encoding cyclin dependent kinase inhibitor 2A), *FAT1* (encoding FAT atypical cadherin 1), and *KMT2D* (encoding lysine methyltransferase 2D), with the exception of *NOTCH* (encoding Notch receptor 1), whose mutation frequency is lower in OSCC compared to CSCC. Finally, we describe the differential composition of the tumor microenvironment and how this influences the aggressiveness of each tumor type. Although both OSCC and CSCC tumors are highly infiltrated by immune cells, high levels of tumor-infiltrating lymphocytes (TILs) have been more frequently reported as predictors of better outcomes in OSCC than CSCC. In conclusion, OSCC and CSCC partially share genetic alterations and possess different causal factors triggering their development. The tumor microenvironment plays a key role determining the outcome of the disease.

## 1. Introduction

Squamous cell carcinomas (SCCs) are among the most frequent solid cancers in humans [1]. They represent a major cause of death worldwide. Their incidence is sharply rising, owing to our increased exposure to carcinogens, such as smoking, alcohol consumption, or human papilloma virus (HPV) infection and ultraviolet radiation (UV) [1,2,3,4]. The skin and mucosa of the head and neck areas are the most frequently affected [3,4].

### 1.1. Oral Squamous Cell Carcinomas

Head and neck cancer (HNC) is the seventh most frequent cancer globally, with more than 830,000 new cases diagnosed annually and an association with a high mortality; approximately 50% of patients die of this disease, generating more than 430,000 deaths per year. HNCs comprise a heterogeneous group of malignancies of the upper aerodigestive tract, salivary glands, and thyroid. Most HNCs are squamous cell carcinomas (SCC) (90%) that can arise from the mucosal epithelium of the oral cavity, pharynx, and larynx [1,2,3].

Oral squamous cell carcinoma (OSCC), one of the main types of HNCs, can arise from the surface of the lips, gums, tongue, mouth, and palate, being the ninth leading cause of death in terms of its mortality [1,3,5]. It is considered to be a major global public threat, with about 350,000 new cases diagnosed a year worldwide and an annual mortality of 175,000. OSCCs have a highly variable clinical course, but their overall survival rate is lower than 50% in 5 years, due to the usual late diagnoses in their advanced stages and the poor response to therapy [5,6].

Smoking and alcohol consumption are considered to be major etiologic factors in the development of OSCC [5,7] (Figure 1). Given the decrease in tobacco consumption, this type of cancer is slowly decreasing. However, high-risk human papillomavirus (HPV), primarily HPV-16, and, to a lesser extent, HPV-18 and other types, is an important etiologic agent in oral cancers [7,8]. HPV-OSCC differs from the usual epidemiology of classical OSCC by appearing in younger patients, 10–30 years after their exposure to the virus. It is most commonly a small tumor with cervical adenopathy. HPV-positive OSCC generally has a better prognosis than HPV-negative OSCC [8,9]. This review is focused mainly on HPV-negative OSCC.

OSCC develops from premalignant dysplastic lesions present in the erythroplakia, leukoplakia, and lichen planus, or combinations of these conditions that can progress, leading to invasive cancer [3,7]. However, most patients present with advanced-stage OSCC without a clinical history of pre-malignancy. Several studies using mice have provided evidence that basal cells are the origin of OSCC [3,10]. In addition, genetic factors also contribute to the risk of suffering from OSCC. Individuals with Fanconi anemia, a rare, inherited genetic disease characterized by impaired DNA repair, have a 500-to-700-fold increased risk of developing OSCC [11]. 

OSCC is generally treated with surgical resection, followed by adjuvant radiation or chemotherapy plus radiation, depending on the disease stage [6,12]. In fact, the treatments for most patients with OSCC require multimodality approaches, except for early-stage oral cavity cancers, which are treated with surgery alone. Among the main chemotherapeutic compounds approved by the FDA for different stages of OSCC are cisplatin, docetaxel, bleomycin sulfate, and hydroxyurea. The epidermal growth factor receptor (EGFR; also known as HER1) is overexpressed in up to 90% of HNSCCs and high levels of this protein are correlated with decreased survival [13]. The monoclonal antibody cetuximab used against this protein is approved by the FDA as a radiation-sensitizing agent for patients undergoing primary radiation-based treatment and patients with recurrent or metastatic disease [14]. Other anti-EGFR agents, such as panitumumab, zalutumumab, gefitinib, afatinib, and lapatinib, are being proven. In addition, immunotherapy has a real impact on OSCC and the immune checkpoint inhibitors, pembrolizumab and nivolumab, are approved for the treatment of cisplatin-refractory recurrent or metastatic OSCC. Additionally, pembrolizumab is approved as first-line therapy for patients who present with unresectable or metastatic disease [15,16].

### 1.2. Cutaneous Squamous Cell Carcinoma

Cutaneous skin cancers are classified into melanoma and non-melanoma skin cancer (NMSC). NMSC is the most frequent cancer worldwide, displaying an annual rate of 5.8% in the world population. It includes basal cell carcinoma (BCC) and cutaneous squamous cell carcinoma (CSCC) as its major subtypes. The incidence of NMSC has been increasing by 3–8% annually and, currently, 2–3 million cases occur per year globally [1,2,4]. In fact, 20% of the population is expected to develop NMSC in their lifetime. Therefore, NMSC represents high economic costs, constituting a severe public health problem [2,4,17].

CSCC, the second most common type of NMSC worldwide, is considered to be highly curable, since it rarely metastasizes (5%). However, metastasis is associated with a poor prognosis and is responsible for most deaths associated with NMSC, with a patient survival rate of 10–20% over 10 years [4,17,18,19]. Despite its global incidence, NMSC is not clearly registered; in the United States, it affects over 700,000 new patients per year [1,2]. Actinic keratosis (AK) is a common skin lesion that is often defined as premalignant, although increasing evidence points to it as an early stage of CSCC. The majority of AKs are reported in middle-aged or elderly individuals, with an estimated incidence of 60% in patients over 40 years of age. It is an erythematous lesion frequently associated with field cancerization, which contains keratinocytes bearing characteristic molecular alterations [1,2,4]. It is estimated that 5-to-20% of AKs will transform into CSCC [18,19]. 

On the other hand, Bowen disease (BD) is defined as an in situ variant representing an early stage of CSCC. Regarding its malignant potential, approximately 3% of BD cases evolve into invasive CSCC [4,17]. In any case, due to the high incidence of CSCC and its related premalignant lesions, its treatment costs are really high and are likely to keep rising with the continuous increase in life expectancy. Additionally, as aggressive or highly invasive CSCC normally occurs in exposed areas such as the ears or nose, surgery has an important negative impact with functional, cosmetic, and psychosocial consequences [20]. 

The pathogenesis of CSCC is multifactorial. Sun exposure, chronic wounds, and immunosuppression are the major risk factors of CSCC [4] (Figure 1). Solar radiation is considered as the main carcinogen responsible for CSCC, since it induces DNA adducts such as cyclobutene pyridine dimers and pyrimidine 6–4 pyrimidone photoproducts, causing DNA mutations and acting as both an initiator and promoter of CSCC. UV radiation is responsible for 95% of keratinocytic cancers [4,18,19] Occasionally, in addition to visible lesions, UV light induces subclinical lesions adjacent to tumor tissues, which has led to the use of the concept “field cancerization”. Apart from UV radiation, inherited genetic conditions, such as Xeroderma Pigmentosum, albinism, or epidermolysis bullosa, are also recognized as predisposing factors in the development of CSCC [18,19].

Surgery is the gold standard treatment for CSCC with various modalities: standard excision, Mohs micrographic surgery, curettage, and electrodessication or cryosurgery [20]. It is commonly accompanied by radiotherapy and/or chemotherapy in patients with high-risk tumors. The interest in non-invasive treatments such as Diclofenac, Imiquimod, and Photodynamic Therapy has grown in recent years [21,22]. As in the case of OSCC, EGFR inhibitors are also used for CSCC. In particular, Cetuximab has been developed and tested on high-risk CSCC patients in clinical trials. Additionally, Cemiplimab, a human monoclonal antibody directed against programmed death 1 protein (PD-1), has been approved by the FDA for patients with locally advanced or metastatic CSCC [22,23]. 

## 2. The Process of Carcinogenesis: Gene Mutations in Oral and Cutaneous Squamous Cell Carcinomas

Oral and cutaneous carcinogenesis are considered to be multi-step and multifocal processes reflecting genetic alterations that result in the transformation of normal cells that undergo premalignant states and eventually become highly malignant [2,10,24,25].

These changes are due to exposition to external agents (physician and biological), giving them advantages for growth and leading to a progressive conversion into neoplastic cells. The carcinogenesis process in SCC consists of three main stages: initiation, promotion, and progression (Figure 1). In the initiation step, an adult stem cell or progenitor cell, located in the basal cell layer of the epithelium, acquires genetic alterations. These cells subsequently proliferate and give rise to a cell patch formed by cells that share genetic alterations, defined as a clonal unit. The promotion process is a change in the patch into an expanding group of cells with additional heterogeneous genetic alterations, but this change is not always clonal expansion [24,25]. Several mutated patches surrounded by normal tissues are considered to be a cancerization field. Initially, the molecular changes in the field of cancerization are not clinically visible. The alterations become irreversible, initially leading to the development of preneoplastic lesions (leukoplasia in OSCC and AKs in CSCC). Tumor progression continues with the accumulation of genetic mutations that lead to the conversion of preneoplastic lesions into SCCs. The promotion and progression steps are accompanied by a series of histopathological changes, from cellular atypia through various degrees of dysplasia to carcinoma in situ, before the development of invasive SCC [24,25]. 

Genetic alterations during OSCC and CSCC are cumulative over periods of chronic exposure to carcinogens and affect the tumor suppressor genes and oncogenes. The use of next-generation sequencing and whole-exome sequencing (WES) has allowed for the identification of the driver mutations (those that confer a selective growth advantage to the cell) and passenger mutations (those that have no effect on the selective growth advantage of the cell) responsible for tumor initiation and progression in SCCs [2,4,26].

SCC is characterized by very high background mutation rates, as well as a high genetic instability, with a frequent loss and gain of chromosomal regions. This is particularly evident in the case of CSCC, which is associated with UV damage and presents DNA alterations between 5 to 15 times greater than those of non-cutaneous tumors. With so many mutations, it is complicated to separate the driver mutations from the passenger mutations (those that have no direct or indirect effect on the selective growth advantage of the cell) in CSCC [26,27,28,29,30,31]. It is hypothesized that most of these mutations are “passengers”, with little or no impact on the tumor progression, while a subset represents “driver mutations”, which promote tumorigenesis by regulating the cell fate, growth, survival, or genomic maintenance. 

The significantly mutated genes in HNSCC and CSCC obtained from the Cancer Genome Atlas (TCGA), cBioPortal database, and several publications are summarized in Table 1 [26,27,28,29,30,31,32]. Additionally, data obtained from the COSMIC database are shown in Appendix A. To allow for a global assessment, the HNSCC data are not stratified regarding location (OSCC samples are included within the HNSCC category) or HPV status. Some of the mutated CSCC driver genes include *TP53* (encoding tumor protein p53), *CDKN2A* (encoding cyclin dependent kinase inhibitor 2A), *FAT1* (encoding FAT atypical cadherin 1), NOTCH1-3 (encoding Notch receptor 1-3), *PTEN* (encoding phosphatase and tensin homolog), *KMT2C* (encoding lysine methyltransferase 2C), *PIK3CA* (encoding the catalytic subunit of phosphatidylinosito 3-kinase), HRAS (encoding HRas proto-oncogene), *KNSTRN* (encoding kinetochore localized astrin binding protein), *EGFR* (encoding epidermal growth factor receptor), *CARD11* (encoding caspase recruitment domain family member 1), *MYC* (encoding MYC proto-oncogene) *MAP3K9* (encoding mitogen-activated protein kinase kinase kinase 9), and *NSD2* (encoding nuclear receptor binding SET domain protein ). In the case of OSCC-associated mutations, which have been described to be significantly enriched in tumor suppressor genes, these include *TP53*, *CDKN2A*, *FAT1*, *NOTCH1*, *PTEN*, *KMT2D*, *NSD1*, and *TGFBR2* (encoding transforming growth factor beta receptor 2).

The tumor suppressor gene *TP53*, a critical regulator involved in key cellular activities including DNA repair, cell cycle arrest, and apoptosis, is the most frequently mutated gene in OSCC and CSCC. Its alteration occurs early in the progression from normal to precancerous tissue [26,28,29,30,31,32,33,34,35,36]. In fact, mutations in this gene are considered to be a UV signature in CSCC [28,29,30,31,32]. In addition, mutations in *CDKN2A* have been described in both OSCC and CSCC, with a relevant role in the progression of AK to CSCC [26,28,29,30,31,32,33,34,35,36,37,38,39].

One important signaling cascade implicated in regulating the cell fate decisions in the development and homeostasis of oral and skin cancers is the NOTCH signaling pathway [40,41,42,43]. In fact, NOTCH1 mutations are considered to be early events in squamous carcinogenesis of the skin. Their loss is associated with disease progression, being mutated in 75% of CSCCs [42]. In contrast, mutations in the *NOTCH* genes are found to a lesser extent in OSCCs (~18%) [40].

Another tumor suppressor gene typically mutated in both OSCC and CSCC is *FAT1* (FAT atypical cadherin), a member of the cadherin superfamily. The inactivation of FAT1 results in aberrant Wnt/β-catenin signaling that promotes tumorigenesis. FAT1 is altered in 23% of HNSCCs [33,43] and 46–70% of CSCCs [28,29]. In CSCC, a loss in function of FAT1 promotes tumor initiation, progression, invasiveness, stemness, and metastasis through the induction of a hybrid EMT state [44]. In HNSCC, including in OSCC cell lines, it has been proposed that YAP1 may represent an attractive precision therapeutic option for cancers harboring genomic alterations in the tumor suppressor gene *FAT1* [45].

TP63 plays a critical role in epithelial development and homeostasis. It operates as a negative regulator of NOTCH1, controlling the switch between proliferation and differentiation, and can act as an oncogene, as it is frequently overexpressed in OSCC and CSCC [30,32,43]. Additionally, mutations in the oncogenes implicated in the PI3K/AKT/mTOR and MAPK/ERK signaling pathways have been described in OSCC and CSCC. The PI3K/AKT/mTOR pathway is the most frequently altered oncogenic pathway in HNSCC [15,34,43]. In fact, the *PIK3CA* gene is mutated in over one third of OSCCs [3,46]. The RAS pathway is activated in the malignant progression of CSCC [47]. Gain-of-function mutations in *HRAS* have been identified in up to 23% of CSCCs. However, mutations in these pathways in OSCC are infrequent, with HRAS mutations being the most common (~4% of tumors) [43]. Additionally, mutations in the epidermal growth factor receptor (*EGFR*) are found in these types of SCCs [29,47]. While EGFR is overexpressed in 80–90% of HNSCC tumors and associated with poor overall survival, its mutation rate is reported as being limited to 1–20% in CSCC [30,48].

Finally, mutations in chromatin-modifying enzymes, leading to DNA and histone modifications, have been described in cancer. In this sense, loss-of-function mutations in the H3K4 methyltransferases, KMT2C, and KMT2D have been described in CSCC [49]. Additionally, inactivating mutations have been reported in the lysine demethylase 6A (KDM6A) in HNSCC [27,28,29].

## 3. Tumor Microenvironment

SCC and its precursor lesions are complex systems in which neoplastic cells coexist with the other cell types and tissue components that comprise the tumor microenvironment (TME). The TME contains multiple different cell types, including cancer-associated fibroblasts (CAFs), neutrophils, macrophages, and regulatory T cells (Figure 2). Tumor cells and TME cell populations interact with each other via complex communication networks through the various secreted cytokines, chemokines, growth factors, and proteins of the extracellular matrix (ECM). The TME is known to be implicated in cancer cell survival, tumor progression, and the tumor response to therapy [50].

### 3.1. Extracellular Matrix

The extracellular matrix (ECM) is a non-cellular network of macromolecules (collagen, fibronectin, and laminin, etc.) that offers structural and biochemical support for cellular components, enabling it to influence cell communication, adhesion, and proliferation [51]. In cancer, the ECM is frequently deregulated and disorganized, which directly stimulates malignant cell transformation. Matrix metalloproteinases (MMPs) are critical molecules for the EMT process because they not only degrade cell adhesion molecules, favoring migration and metastasis, but also promote the initiation and proliferation of primary tumors [52]. MMPS are produced by tumor and stromal cells, such as fibroblast and inflammatory cells [48,51,52]. Alterations in the proteins of the ECM may impact the tumor progression in SCCs. Collagen is the major protein component of the ECM, which provides the cells with tensile strength and support for migration [51]. The loss of type IV collagen correlates with poorly differentiated OSCC [52] and CSCC [53]. Fibronectin is produced by fibroblasts and endothelial cells and mediates the cellular interaction with the ECM. In the development of cancer, increased levels of fibronectin have been associated with increased tumor progression, migration, and invasion, as well as an impaired response to treatment [54,55]. Additionally, the expression of the laminin receptor plays an important role in SCC progression [55,56], as reduced laminin expression has been correlated with an invasive phenotype of OSCC tumors [53].

### 3.2. Cancer Associated Fibroblasts

Fibroblasts are one of the main cell components of the connective tissue subjacent to the epithelia. The main functions of these cells are the synthesis of the ECM (collagen, laminin, and fibronectin, including those needed to form basal membranes), the regulation of epithelial differentiation, and the promotion of wound closure [57,58]. CAFs are activated fibroblasts with mesenchymal characteristics associated with cancer cells, which contribute to tumor-promoting inflammation and fibrosis. CAFs acquire specific characteristics, such as a distinct morphology (an elongated spindle-like shape), and express differential markers (α-sma—Alpha-Smooth Muscle Actin; FAP-1—Fibroblast Activation Protein-1; vimentin and S100A4) and a lack of lineage markers for epithelial cells, endothelial cells, and hematopoietic cells [59,60] (Figure 2). However, the precise origins and roles of the fibroblast populations within the tumor microenvironment remain poorly understood.

In the case of HNSCC, several populations of CAFs have been described that, according to specific markers, can be classified into three subgroups: classical CAFs, normal activated fibroblasts, and elastic fibroblasts. Classical CAFs are enriched for genes, encoding proteins such as FAP, PDGF (platelet-derived growth factor receptor), lysyl oxidase, and MMPs. Normal activated fibroblasts show a low expression of CAF markers and elastic fibroblasts are enriched for tropoelastin, fibrillin 1, and microfibril-associated protein 4. It seems that the presence of different CAFs can be related to overall patient prognoses [61].

CAFs are key in different stages of the development of OSSC and CSCC, stimulating the growth and progression of tumors and participating in the maintenance of a state of poor differentiation of the surrounding cells. They act synergistically with epithelial cells to promote carcinogenesis and influence the patterns of invasiveness and metastasis [61,62]. In this sense, CAFs could act in the initiation process of cancer, favoring the mutagenicity of epithelial cells, for example, by secreting ROS, which favor decreases in the pH and hypoxia in the TEM [63]. In tumor progression, they promote the migration and invasion of tumor cells by chronically maintaining the proinflammatory stimuli via the promotion of oxidative stress. They also contribute to this end by secreting a broad amount of cytokines and chemokines, such as transforming growth factor beta (TGFβ) and interleukins (IL-1 and IL-6), and a broad range of growth factors, such as EGF (epidermal growth factor), bFGF (basic fibroblast growth factor), VEGF (vascular endothelial growth factor), HGF (hepatocyte growth factor), tumor necrosis factor (TNF), interferon-(IFN), CXCL12, IL-6, galectin-1, sonic hedgehog protein (SHH), and bone morphogenetic protein (BMP), among others, which are tumor-promoting. In particular, HGF has been described to promote glycolysis in HNSCC cells [63]. All these molecules can influence tumor cell growth, angiogenesis, and the recruitment of immunosuppressive immune cells [64,65,66]. CAFs are also crucial producers of MMPs, playing an important role in modulating the TME through the remodeling and degradation of the ECM, which ultimately results in the promotion of the invasive phenotype of cancer cells [66,67,68].

In addition, they promote the EMT by secreting a variety of soluble activators that initiate the TGFβ cascade in epithelial tumor cells, leading to a change in morphology and response, acquiring mesenchymal characteristics [69,70]. As a major secreted factor of CAFs, TGFβ predominantly mediates the crosstalk between CAFs and cancer cells. Several in vitro studies have demonstrated that, in OSCC and CSCC, the TGFβ secreted by CAFs induces EMT and resistance to different therapies [71,72,73,74,75].

### 3.3. Immune Cells

The immune cell component of the TME is formed by tumor-infiltrating lymphocytes (TILs, including CD4+ and CD8+ T cells, B cells, natural killer T cells, and myeloid lineage cells (macrophages, neutrophils, monocytes, eosinophils, myeloid-derived suppressor cells or MDSCs, and mast cells or MCs)). In general, OSCC and CSCC tumors are highly infiltrated by immune cells, although the extent and composition of the immune cell infiltrate vary according to the anatomical subsite and etiological agent [76,77] (Figure 2).

Tumor-infiltrating lymphocytes (TILs) are the major cell type in adaptive immunity that recognize specific antigens and produce specific immune responses. High levels of TILs generally correspond to better outcomes in OSCC, but this is dependent on the balance of cells with anti-tumor activity (effector T or Teff cells) versus those with immunosuppressive activity (regulatory T or Treg cells) in the TIL population [76,78].Teffs are related to high levels of cytotoxic CD8+, which produces IFN-ϒ [76]. On the contrary, high levels of CD4+Foxp3+ or Treg cells through IL-4 and IL-10 production have been correlated with immunosuppression and pro-tumor activity, triggering poor outcomes [79,80]. Likewise, UV radiation also causes an increase in Treg and a decrease in Teff cells in the skin, leading to a change in the T-cell balance and promoting the development of CSCC [81]. On the other hand, the presence of B lymphocytes is related to higher levels of CD8+ cell infiltration and, therefore, to a better prognosis in OSCC and CSCC [82,83].

Tertiary lymphoid structures (TLS) are crucial elements of the tumor immune microenvironment, corresponding to sites of lymphoid neogenesis with the potential of orchestrating anti-tumor responses. They correspond to ectopic lymphoid organs, emerging in the context of chronic inflammation such as the TME and even allowing for germinal center formation [84,85,86]. In OSCC patients, a high density of TLS has been associated with a better overall survival and identified as an independent positive prognostic factor [87,88]. In the case of CSCC, though not much work has been performed, clinically, the presence of TLS has been prominently associated with a better degree of histopathological grades and a higher level of sun exposure. Furthermore, the presence of intratumoral TLS has been associated with lower lymphovascular invasion. Therefore, TLSs are considered to be a positive prognostic factor for CSCC and will provide a theoretical basis for the future diagnostic and therapeutic value in this type of cancer [89]. The features and clinical significance of TLSs in SCC still remain unknown.

Tumor-associated macrophages (TAMs) interact, modulate, and influence tumor progression, invasion, and metastasis. Macrophages display a great plasticity, oscillating between M1 (antitumoral) and M2 (protumoral) phenotypes. M1 macrophages produce pro-inflammatory cytokines (IL-12 and IL-23), tumor necrosis factor-α (TNF-α), and chemokines (CCL-5, CXCL9, CXCL10, and CXCL5), which promote adaptive immunity. They also express high levels of major histocompatibility complex 2 (MHC-2) molecules, allowing for the presentation of tumor antigens [90,91]. In contrast, M2 macrophages play an immunoregulatory role and are involved in tissue remodeling, angiogenesis, and tumor progression. M2 macrophages act by releasing anti-inflammatory cytokines (IL-4, IL-13, IL-10, and TGFβ, etc.), overexpressing PD-L1 (Programmed death-ligand 1) and expressing comparatively lower levels of MHC-2 molecules [90,91,92]. Several studies have suggested a correlation between the level of TAM infiltration and a poor outcome in OSCC, which could be used as a potential prognostic marker [93,94].

Myeloid-derived suppressor cells (MDCSs) comprise a heterogeneous population of cells that play a crucial role in the negative regulation of the immune response in cancer by inhibiting both adaptive and innate immunity, establishing the premetastatic niche in different types of cancer [95,96]. In addition, MDSCs have also been linked to angiogenesis and the degradation of the ECM [97,98]. A high abundance of circulating MDSCs correlates with advanced stages of OSCC and is also known to promote CSCC development [98,99,100].

Mast cells (MCs) represent another important myeloid component of the immune system. MCs in the TME may have pro-tumoral functions, such as the promotion of angiogenesis (through VEGF production), ECM degradation (via MMPs production), and the induction of tumor cell proliferation (through tryptase and histamine) [101,102]. In OSCC and CSCC, the protective and pro-tumoral role of MCs has been described in several studies [103,104,105].

### 3.4. The Importance of the TME for the Treatment of OSCC and CSCC

OSCC and CSCC are generally treated with surgical resection and, depending on the disease state, this is accompanied by radiation or chemotherapy [6,12,20]. Traditional tumor treatment methods cannot solve the problems of tumor recurrence and metastasis, which are often associated with the TME, as mentioned in the previous section. Therefore, new single or combined strategies are being developed to address the TME, as described below.

#### 3.4.1. CAF-Targeting Strategies

The importance of CAFs in tumor development and their role in therapy resistance have been demonstrated in several types of cancer, including OSCC and CSCC [64]. CAF-mediated resistance to cetuximab has been reported in OSCC [106,107]. Additionally, in CSCC, the presence of CAFS has been found to increase resistance to photodynamic therapy [71], suggesting that therapeutic CAF targeting could increase the response rates for a diverse range of treatments.

One of the main mediators related to these resistance effects is TGFβ. This cytokine modulates the tumor progression and therapy response through the CAF activation status, shape, and invasiveness [108]. Quan et al. [109] suggested that TGF-β1 induces EMT to increase the capacity of OSCC for invasion, and Gallego et al. [71] described that an increased secretion of CAF-derived TFGβ mediates resistance in CSCC. However, targeting TGF-β is potentially problematic for its dual role: in the early stages of tumorigenesis it can act as a tumor suppressor, while acting as a tumor promoter in later stages [110]. Even so, it has been described that a novel TGFβ inhibitor promoted anti-tumor immune responses in OSCC, alone and in combination with anti PD-L1 antibodies [111]. Although the signaling cascades involving TGFβ are the primary signaling pathways regulating CAF activation, there are other growth factors and signaling molecules also implicated in the differentiation process, including NOX4 (NADPH oxidase 4), FGF, IL-6, or TNF [112]. Hanley et al. [113] identified NOX4 as a critical regulator of CAF activation in OSCC, and its inhibitor Setanaxib suppressed CAF activation. In OSCCs and CSCCs, there have not been many more studies targeting CAFS in order to prevent resistance. However, in other cancers, more trials have been described, such as the depletion of FAP-expressing cells as an adjuvant to immunotherapy [114]. Additionally, a combination of paclitaxel (which suppresses the expression of α-SMA) with gemcitabine improved the overall survival in pancreatic cancer patients [115], suggesting a possible new therapeutic window for OSCC and CSCC.

#### 3.4.2. Immunotherapy

Immunotherapy has outstanding application value in the field of tumor therapy, including antibody-based therapy, cytokine therapy, and gene therapy.

One of the main strategies for eliminating SCCs is based on the use of monoclonal antibodies (mAbs), which include immune checkpoint inhibitors or anti-angiogenesis mAbs.

There are several clinical trials focused on mAbs for OSCC and CSCC treatment (Table 2).

PD-1 is a checkpoint protein belonging to a group of T cell receptors involved in T cell suppression. PD-1 is also expressed by B cells, monocytes, and natural killer and dendritic cells [116]. This transmembrane protein binds to PD-L1, which is present on the surface of tumor cells, and this interaction triggers a signal that inhibits the activated T cells and induces immunological exhaustion and T cell apoptosis [116,117]. Then, the PD-L1/PD-1 axis is a primary mechanism of cancer immune evasion and has thus become the main target for the development new drugs that have emerged in recent years. Targeting the immune checkpoint proteins with mAbs has yielded a net clinical benefit in cancer [118,119]. So far, several mAbs have been approved for PD-1/PD-L1 blockade in clinical studies for oral cancer treatment, including Cemiplimab, Nivolumab, Sintilimab, Toripalimab, Pembrolizumab, Aezolizumab, Avelumab, Camreluzimab, and Durvalamab (Table 2). Likewise, mAbs blocking other immune checkpoint receptors such as CTLA-4 (Tremelimumab) are being studied [120]. Finally, in OSCC, the effect of inhibiting OX40, a costimulatory molecule that can enhance T cell immunity, is also being tested. Anti-human OX40 was used in a phase I clinical trial (NCT02274155) prior to surgery. The results demonstrated that anti-OX40 mAb could induce the activation and proliferation of T cells in hosts, suggesting its successful potential as a clinical strategy [121].

On the other hand, as angiogenesis plays an important role in tumor development and metastasis, mAbs are also being tested to inhibit these processes in both OSCC and CSCC (Table 2). Cetuximab, an anti-EGFR mAb, and bevacizumab, an anti-VEGFR mAb, are being administered in clinical trials, either alone or in combination with other treatments.

Macrophages and fibroblasts, among other cell types belonging to the TME, produce different cytokines that can be pro- or anti-tumorigenic [73,90,91]. Several cytokines clinical trials have been completed in OSCC, although the results have not been published yet. Some of these trials are related to the administration of IL-2 (NCT00899821 and NCT00019331) or INFα (NCT00276523, NCT00054561, NCT00002506, and NCT00014261), in order to see if they promote anti-tumor responses in combination with other treatments.

## 4. Conclusions and Future Directions

The data published from the different cohorts that have been evaluated up to now do not show striking differences between the genes implicated in the genesis of OSCC and CSCC. Driver mutations in tumor suppressor genes seem to be more frequent in OSCC, including TP53, CDKN2, FAT1, and KMT2D, with the exception of NOTCH, which is mutated with less frequency compared to CSCC. TP53 deletion is the most frequent tumor suppressor gene mutated in human SCCs and this alteration is usually sufficient for transforming benign tumors into malignant CSCCs when another oncogene (RAS or CCND1) is also mutated, as it occurs in many other solid tumor malignancies. In contrast, in the case of OSCC, PIK3CA, encoding the catalytic subunit of PI3K, is the only oncogene found to be frequently mutated. Next-generation sequencing techniques will allow for the identification of new drivers in both types of SCC. Moreover, a better understanding of the transcriptional and chromatin landscapes of OSCC and CSCC will allow for tissue-specific determinants that regulate the tumor initiation and progression of OSCC and CSCC.

In OSCC and CSCC, neoplastic cells coexist with the TME, which is implicated in cancer cell survival, tumor progression, and responses to therapy. Changes in the ECM components, such as reduction in type IV collagen or laminin, correlate with poorly differentiated OSCC and CSCC. Additionally, CAFs are key in different stages of the development of OSSC and CSCC, participating in maintaining a state of poor differentiation of the surrounding cells. Several studies have demonstrated that TGFβ secreted by CAFs induces EMT and resistance to different therapies. However, there have not been many more studies targeting CAFs to prevent resistance in OSCC and CSCC. This is therefore a possible therapeutic window to be studied. In general, OSCC and CSCC tumors are highly infiltrated by immune cells. High levels of TILs generally correspond to better outcomes in OSCC, but this is dependent on the balance of cells with anti-tumor activity versus those with immunosuppressive activity. Today, the use of monoclonal antibodies is being tested in several clinical trials as a promising therapeutic strategy, with the aim of eliminating SCCs.

## Figures and Tables

**Figure 1 cancers-15-03227-f001:**
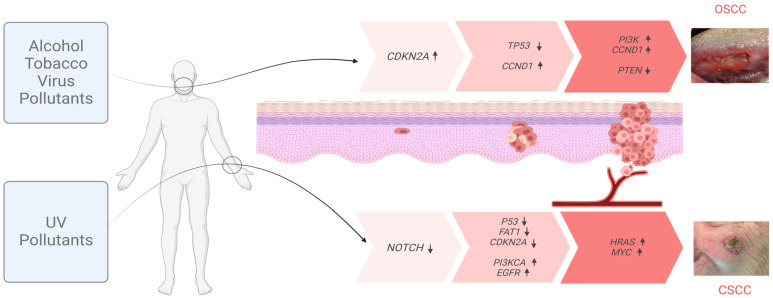
Inducing factors and multistep progression of OSCC and CSCC. Mutagenic agents responsible for the initiation and progression of OSCC (**top**) and CSCC (**bottom**) tumors. The main genes bearing characteristic cumulative mutations at each step of the process are indicated. Arrows next to each gene indicate the changes in the expression (↑ up; ↓ down).

**Figure 2 cancers-15-03227-f002:**
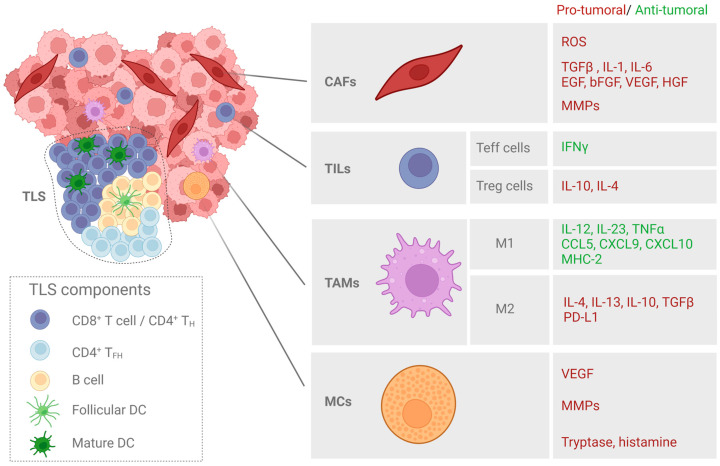
Major cellular components of the tumor stroma in SCC. Cancer-associated fibroblasts (CAFs), tumor-infiltrated lymphocytes (TILs), tumor-associated macrophages (TAMs), and mast cells (MCs) are among the major cellular components of the tumor stroma. The molecules with anti- or pro-tumoral effects secreted by each tumor-associated cell population are indicated. Tertiary lymphoid structures (TLSs) are also highlighted as areas enriched in T and B cells that play a role in facilitating the influx of immune cells into the tumor site.

**Table 1 cancers-15-03227-t001:** Significantly mutated genes in HNSCC and CSCC.

Type of Gene	Gene	Protein	Function	Mutation Frequency (%) HNSCC/CSCC
Tumor suppressors	* TCP53 *	p53	Cell survival and proliferation	68/76
	* CDKN2A *	p16 INK4A	Cell cycle and survival	20/34
	* PTEN *	PTEN	Cell cycle	3.1/5.4
	* FAT1 *	Protocadherin FAT1	Cell adhesion (Wnt/β-catenin pathway)	20.3/55.5
	* AJUBA *	AJUBA	Cell adhesion (Wnt/β-catenin pathway)	6.4/10.7
	* NOTCH (1) *	NOTCH (1)	Cell adhesion and differentiation (Delta/Notch pathway)	16.3/55
	* KMT2D *	Histone-lysine N-methyltransferase KMT2D	Epigenetic regulation	14.5/41
	* LRP1B *	LRP1	(Wnt/β-catenin pathway)	16.7/56.3
Oncogenes	* HRAS *	HRAS	Cell proliferation (MAP/ERK pathway)	6/12.6
	* PI3KCA *	p110	Cell proliferation (PI3K/AKT pathway)	15.5/13.2
	* EGFR *	EGFR	Cell proliferation (PI3K/AKT and MAP/ERK pathway)	3.7/8.6
	* TP63 *	p63	Cell cycle	2.9/9.3
	* CCND1 *	G1–S-specific cyclin D1	Cell proliferation	0.6/2.9
	* TGFBR2 *	TGFBR		4.6/5.7

Mutation percentage obtained in the cBioPortal data base (https://www.cbioportal.org; accessed on 28 April 2023) (686 samples of HNSCC and OSCC, excluding nasopharingeal and salivary carcinomas, and 151 samples of CSCC. *TP53* (encoding p53 protein), *CDKN2A* (encoding cyclin dependent kinase inhibitor 2A), *PTEN* (encoding phosphatase and tensin homolog), *FAT1* (encoding FAT atypical cadherin 1), *AJUBA* (encoding ajuba LIM protein), *NOTCH1* (encoding Notch receptor 1-3), *KMT2C* (encoding lysine methyltransferase 2C), *LRP1B* (LDL receptor related protein 1B), HRAS (encoding HRas proto-oncogene), *PIK3CA* (encoding the catalytic subunit of phosphatidylinosito 3-kinase), *EGFR* (encoding epidermal growth factor receptor), *TP63* (encoding tp63 protein, CCND1 (encoding cyclin D1), *TGFBR2* (encoding transforming growth factor beta receptor 2).

**Table 2 cancers-15-03227-t002:** Monoclonal antibody-based trials for oral and cutaneous squamous cell carcinomas.

	Product Name	Target	Phase	Status	Identifier
OSCC	Anti-EGFR monoclonal antibody	EGFR	II	Recruiting	NCT04508829
	Cetuximab	EGFR	II	Completed	NCT00933387
	Anti-OX40 antibody	OX40	I	Active, not recruiting	NCT02274155
	Cemiplimab	PD-1	II	Recruiting	NCT04398524
	Nivolumab	PD-1	I/II	Completed	NCT03021993
	Sintilimab	PD-1	II	Not yet recruiting	NCT05000892
	Toripalimab	PD-1	II/III	Recruiting	NCT05125055
	Camrelizumab + Apatinib	PD-1 + VEGFR	II	Recruiting	NCT05069857
	Durvalumab + Tremelimumab	PD-L1 + CTLA-4	II	Active, not recruiting	NCT03410615
	Bevacizumab	VEGF	I	Active, not recruiting	NCT01552434
CSCC	Erlotinib	EGFR	II	Completed	NCT01059305
	Cemiplimab	PD-1	II	Completed	NCT04242173
	Cemiplimab	PD-1	I	Recruiting	NCT03889912
	Cemiplimab	PD-1	I	Recruiting	NCT04339062
	Pembrolizumab	PD-1	II	Recruiting	NCT04808999
	Pembrolizumab	PD-1	II	Active, not recruiting	NCT03284424
	Pembrolizumab	PD-1	III	Recruiting	NCT03833167
	Pembrolizumab + Cetuximab	PD-1 + EGFR	II	Unknown	NCT03666325
	Atezolizumab	PD-L1	I	Recruiting	NCT05085496
	Avelumab + Cetuximab	PDL-1 + EGFR	II	Recruiting	NCT03944941

Data obtained from the following website (https://clinicaltrials.gov/ct2/home, accessed on 28 April 2023) PD-1, Programmed cell death-1; VEGF-A, Vascular endothelial growth factor A; EGFR, Endothelial growth factor receptor; OX40 (CD134), A member of the tumor necrosis factor family of receptors; CTLA-4, Cytotoxic T-lymphocyte-associated protein 4; and PD-L1, Programmed death-ligand 1.

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
