# Peer review of "The Distinctive Features behind the Aggressiveness of Oral and Cutaneous Squamous Cell Carcinomas"

_cancers, 2023, doi:10.3390/cancers15123227_

Round 1
Reviewer 1 Report
The manuscript by Alonso-Juarranz and collaborators is well-written and absolutely readable by people not in the field. This work proposes a summary of the main characteristics of OSCC and CSCC, which in my opinion are important to underline, especially when in vitro and molecular investigations are being made.
Author Response
The manuscript by Alonso-Juarranz and collaborators is well-written and absolutely readable by people not in the field. This work proposes a summary of the main characteristics of OSCC and CSCC, which in my opinion are important to underline, especially when in vitro and molecular investigations are being made.
We thank the reviewer the positive appreciation on the manuscript.
Reviewer 2 Report
The manuscript summarizes basic concepts of mutational landscape and the tumor microenvironment driving cutaneous and head and neck squamous cell carcinomas, and its correlation with patient prognosis or treatment response. Overall, the paper is easy to follow and gives a general idea of the diseases. Although, in an attempt to talk about two different cancers at the time, the authors fail to describe specific information or very recent discoveries in both tumor types and as a result it may be too general.
Specific comments:
Table1: TP53 is spelled wrong.
Major histocompatibility complex 2 (MHC-2) is wrong spelled in lines 342 and 347.
The authors are using TCGA and cBioportal information as independent sources of information, but cBioportal contains the TCGA data set as well as other cohorts including the firehouse legacy and TCGA PanCancer altas, which includes the published TCGA samples, so they are redundant. The authors should distinguish datasets to don’t make misleading conclusions based on duplicated genomic data. In addition, mutations are described for head and neck cancers, and it is not clear if they are distinguishing by location (oral or others) and HPV status.
Table 1 with the mutations is unclear: there is no TCGA dataset for CSCC, what are the numbers in the TCGA column? For example, for TP53: 41-72/42-95, what are these numbers and intervals?
The statement “The inactivation of FAT1 results in aberrant Wnt/β-catenin signaling that promotes tumorigenesis” is an oversimplification of the function of FAT1 that has been described at least to control cell plasticity and hybrid EMT state in CSCC models (DOI: 10.1038/s41586-020-03046-1), and Hippo pathway in HNSCCs (DOI: 10.1038/s41467-018-04590-1). The authors should cite this important information and cite the paper regarding FAT1 function in Wnt pathway.
The authors should really check their citations to make sure the readers can identify the original papers describing the information mentioned (specific papers, no reviews). Here I number a few examples of citations that are wrong or not related to the information described:
- The function of TP63 is mentioned, but the citation talks about NOTCH function, and not TP63 role in OSCC progression. An adequate citation must be included.
- KMT2D and C are also mentioned, but the referenced papers do not mention KMT functions or its role in OSCCs.
- The loss of collagen type IV in OSCC is mentioned, but reference 49 described the role of integrin aV/b3 in other cancer types.
These are just a few examples, but there are many more citations that are not correct all along the text and the authors must thoroughly check this.
The quality of the English is reasonable, although the abstract could be improved by fixing the narrative and the conclusion.
Author Response
The manuscript summarizes basic concepts of mutational landscape and the tumor microenvironment driving cutaneous and head and neck squamous cell carcinomas, and its correlation with patient prognosis or treatment response. Overall, the paper is easy to follow and gives a general idea of the diseases. Although, in an attempt to talk about two different cancers at the time, the authors fail to describe specific information or very recent discoveries in both tumor types and as a result it may be too general.
We thank you very much your comments. We hope that after incorporating the changes suggested; we have incorporated the latest developments in both carcinomas and we hope that the MS has been improved.
Specific comments:
Table1: TP53 is spelled wrong.
Thank you, we have corrected it.
Major histocompatibility complex 2 (MHC-2) is wrong spelled in lines 342 and 347.
Thank you, we have corrected it.
The authors are using TCGA and cBioportal information as independent sources of information, but cBioportal contains the TCGA data set as well as other cohorts including the firehouse legacy and TCGA PanCancer altas, which includes the published TCGA samples, so they are redundant. The authors should distinguish datasets to don’t make misleading conclusions based on duplicated genomic data. In addition, mutations are described for head and neck cancers, and it is not clear if they are distinguishing by location (oral or others) and HPV status.
That's right, cBioportal includes the TCGA data set. Therefore, in the table, we have decided to keep the data obtained from cBioportal, since it includes a larger number of samples than TCGA.
We have included in the table only the specific data relating to HNSCC (686 samples in total). OSCC samples are included within the HNSCC category, but Nasopharingeal carcinoma or Salivary carcinoma samples are excluded from this table. We have decided to consider bulk data for a more global assessment, so the data are not stratified regarding location or HPVstatus. The text has been modified accordingly:
“To allow a global assessment, the data are not stratified regarding location (OSCC samples are included within the HNSCC category) or HPVstatus”
Table 1 with the mutations is unclear: there is no TCGA dataset for CSCC, what are the numbers in the TCGA column? For example, for TP53: 41-72/42-95, what are these numbers and intervals?
Thank you for your comment, as indicate above, we have removed TCGA data and kept the cBioportal data since the number of samples is higher.
The statement “The inactivation of FAT1 results in aberrant Wnt/β-catenin signaling that promotes tumorigenesis” is an oversimplification of the function of FAT1 that has been described at least to control cell plasticity and hybrid EMT state in CSCC models (DOI: 10.1038/s41586-020-03046-1), and Hippo pathway in HNSCCs (DOI: 10.1038/s41467-018-04590-1). The authors should cite this important information and cite the paper regarding FAT1 function in Wnt pathway.
Thank you very much for your input. We have expanded the paragraph on FAT1, including both articles as follows: “Another tumor suppressor gene typically mutated in both OSCC and CSCC is FAT1 […] [28,29]. In CSCC, the loss of function of FAT1 promotes tumour initiation, progression, invasiveness, stemness and metastasis through the induction of a hybrid EMT state [41]. In HNSCC, in-cluding in OSCC cell lines, it has been proposed that YAP1 may represent an attractive precision therapeutic option for cancers harboring genomic alterations in the tumour suppressor genes FAT1 [42].”
The authors should really check their citations to make sure the readers can identify the original papers describing the information mentioned (specific papers, no reviews). Here I number a few examples of citations that are wrong or not related to the information described:
- The function of TP63 is mentioned, but the citation talks about NOTCH function, and not TP63 role in OSCC progression. An adequate citation must be included.
- KMT2D and C are also mentioned, but the referenced papers do not mention KMT functions or its role in OSCCs.
- The loss of collagen type IV in OSCC is mentioned, but reference 49 described the role of integrin aV/b3 in other cancer types.
These are just a few examples, but there are many more citations that are not correct all along the text and the authors must thoroughly check this.
Thank you very much for your comments. We have checked and corrected the references Accordingly,
The quality of the English is reasonable, although the abstract could be improved by fixing the narrative and the conclusion.
The abstract has been revised and we think that it has been improved.
Reviewer 3 Report
The manuscript entitled “Distinctive features behind the aggressiveness of oral and cutaneous squamous cell carcinomas” by Miguel Alonso-Juarranz et al., was presented as a Literature Review with general focus on the molecular and genetic landscapes underpinning two different types of SCC. Globally, the manuscript was presented in a comprehensive and interesting way, is well-written and provides a good overview of the actual knowledge regarding the pathogenesis of oral and cutaneous SCC. The Introduction section is coherent, well-composed and documented, and it sets the context the data that has been reviewed. Some paragraphs need to be extended adding newest recent findings. It would be necessary to better develop the section “2. The process of carcinogenesis: Gene mutations in Oral and Cutaneous Squamous Cell Carcinomas”. With respect to how mutations in tumor suppressor genes and protooncogenes cooperate to determine the differentiation, aggressiveness, and metastatic potential of OSCC and CSCC, the authors take into consideration to update the bibliography integrating recent studies highlighting this important role of p53 in prevention of tumor development and the maintenance of genome stability just in the SCC context (PMID: 35408839; PMID: 34572732; PMID: 36871014). The figures, being pictures or tables are properly described and annotated. The Conclusions section is clear in formulation but appears too short. Please, improve it perhaps adding a discussion section.
Based on personal proficiency in English, the reviewer retains that a Minor editing of the English language is required.
Author Response
The manuscript entitled “Distinctive features behind the aggressiveness of oral and cutaneous squamous cell carcinomas” by Miguel Alonso-Juarranz et al., was presented as a Literature Review with general focus on the molecular and genetic landscapes underpinning two different types of SCC. Globally, the manuscript was presented in a comprehensive and interesting way, is well-written and provides a good overview of the actual knowledge regarding the pathogenesis of oral and cutaneous SCC. The Introduction section is coherent, well-composed and documented, and it sets the context the data that has been reviewed.
Some paragraphs need to be extended adding newest recent findings. It would be necessary to better develop the section “2. The process of carcinogenesis: Gene mutations in Oral and Cutaneous Squamous Cell Carcinomas”. With respect to how mutations in tumor suppressor genes and protooncogenes cooperate to determine the differentiation, aggressiveness, and metastatic potential of OSCC and CSCC, the authors take into consideration to update the bibliography integrating recent studies highlighting this important role of p53 in prevention of tumor development and the maintenance of genome stability just in the SCC context (PMID: 35408839; PMID: 34572732; PMID: 36871014). The figures, being pictures or tables are properly described and annotated. The Conclusions section is clear in formulation but appears too short. Please, improve it perhaps adding a discussion section.
We thank the reviewer the positive appreciation on the manuscript.
As suggested, in relation to paragraph 2, we have added the following information to expand on the content. “The changes are due to the exposition to external agents (physician and biological) giving them advantages for growth and leading to a progressive conversion into neoplastic cells. The carcinogenesis process in SCC consists of […]. In the initiation step, an adult stem cell or progenitor cell, located in the basal cell layer of the epithelium, acquires genetic alterations. These cells subsequently proliferate and give rise to a cell patch formed by cells that share genetic alterations, which is defined as a clonal unit. The promotion […] expansion [24,25]. Several mutated patches surrounded by normal tissues are considered the cancerization field. Initially, the molecular changes in the field of cancerization are not clinically visible. The alterations become irreversible, […].”
In addition, we have expanded this section with a paragraph on p53 and another on NOTCH, including the bibliography you have mentioned. “The tumor suppressor gene TP53, a critical regulator involved in key cellular activities including DNA repair, cell-cycle arrest and apoptosis, is the most frequently mutated gene in OSCC and CSCC. Its alteration occurs early in the progression from normal to precancerous tissue [15,26,28–36]. In fact, mutations in this gene are considered a UV signature in CSCC [28–32]. In addition, mutations in the inhibitor of cyclin-dependent kinase 2A (CDKN2A) have been described in both OSCC and CSCC, with a relevant role in the progression of AK to CSCC [15,26,28–36].
One important signaling cascade implicated in regulating cell fate decisions in the development and homeostasis of oral and skin cancers is the NOTCH signaling pathway [37-40]. In fact, NOTCH1 mutations are considered an early event in squamous carcinogenesis of the skin. Its loss is associated with disease progression, being mutated in 75% of CSCC [39]. In contrast, mutations in NOTCH genes are found to a lesser extent in OSCCs (~18%) [37].”
The Conclusions section is clear in formulation but appears too short. Please, improve it perhaps adding a discussion section.
Thank you very much for the suggestion. However, as there are no original data, the article itself is constituted as a comparative discussion of existing literature, so we only have the conclusions in a short and concise format.
Round 2
Reviewer 3 Report
The reviewer found the revised version of the manuscript much improved in terms of data presentation and text organization. However, he retains that there are several key recent references missing from the review according to a simple PubMed search, mostly in the genomic landscape of cSCC and p53 part (see below). Thank you for your attention to these newer references.
Mitsui H, et al. Gene expression profiling of the leading edge of cutaneous squamous cell carcinoma: IL-24-driven MMP-7. J. Invest. Dermatol. 2014;134:1418–1427. doi: 10.1038/jid.2013.494.
Inman GJ, et al. The genomic landscape of cutaneous SCC reveals drivers and a novel azathioprine associated mutational signature. Nat. Commun. 2018;9:3667. doi: 10.1038/s41467-018-06027-1.
Steffens Reinhardt L, et al. The role of truncated p53 isoforms in the DNA damage response. Biochim Biophys Acta Rev Cancer. 2023 doi: 10.1016/j.bbcan.2023.188882.
Nappi A, et al. Loss of p53 activates thyroid hormone via type 2 deiodinase and enhances DNA damage. Nat. Commun. 2023; 14: 1244. doi: 10.1038/s41467-023-36755-y.
Author Response
The reviewer found the revised version of the manuscript much improved in terms of data presentation and text organization. However, he retains that there are several key recent references missing from the review according to a simple PubMed search, mostly in the genomic landscape of cSCC and p53 part (see below). Thank you for your attention to these newer references.
Mitsui H, et al. Gene expression profiling of the leading edge of cutaneous squamous cell carcinoma: IL-24-driven MMP-7. J. Invest. Dermatol. 2014;134:1418–1427. doi: 10.1038/jid.2013.494.
Inman GJ, et al. The genomic landscape of cutaneous SCC reveals drivers and a novel azathioprine associated mutational signature. Nat. Commun. 2018;9:3667. doi: 10.1038/s41467-018-06027-1.
Steffens Reinhardt L, et al. The role of truncated p53 isoforms in the DNA damage response. Biochim Biophys Acta Rev Cancer. 2023 doi: 10.1016/j.bbcan.2023.188882.
Nappi A, et al. Loss of p53 activates thyroid hormone via type 2 deiodinase and enhances DNA damage. Nat. Commun. 2023; 14: 1244. doi: 10.1038/s41467-023-36755-y.
Thank you very much for your suggestions. One of the references indicated by you was already cited in the MS (see reference number 28: Inman G et al.,2018).
The other three articles (Mitsui et al, 2014; Steffens et al., 2023 and Nappi et al., 2023) have been incorporated with the reference numbers: 37, 38 and 39).
Thank you again for your suggestions.